# OpenReview forum: "Parameter-Free Compressed Federated Learning"
_ICLR.cc/2026/Conference — ICLR 2026 Conference Withdrawn Submission_

### Official Review · Reviewer_qqzp · 2025-10-19

**Soundness:** 2
**Presentation:** 2
**Contribution:** 2
**Rating:** 2
**Confidence:** 4

**Summary:**

This paper proposes two novel algorithms: ParFreFL (Parameter-Free Federated Learning) and ComParFreFL (Compressed Parameter-Free Federated Learning). The core contribution is eliminating hyperparameter tuning in federated learning while significantly reducing communication costs. ParFreFL halves the communication burden of PAdaMFed (a prior parameter-free FL method) by transmitting only one model-sized parameter per round instead of two. ComParFreFL extends this with compression that unifies momentum increment and error feedback, handling biased compression while maintaining parameter-free properties. Notably, ComParFreFL operates independently of compression ratio—claimed as the first such algorithm. Experiments on MNIST and Fashion-MNIST demonstrate competitive or superior performance compared to tuned baselines.

**Strengths:**

# 1. Originality:

The idea of reducing PAdaMFed's communication from $2d$ to $d$ parameters while preserving parameter-free properties is clever. Claim of being "the first instance of such robustness in the compressed FL literature" is interesting if true.

# 2. Quality:

The authors provide rigorous theoretical analysis for theorems 1 and 2 with detailed proofs. It also handles arbitrary data heterogeneity while no bounded dissimilarity assumption.

# 3. Clarity:

The paper is in general well organized, with clear problem formulation and algorithm pseudocode.

# 4. Significance:

This work eliminates hyperparameter tuning, and  communication complexity improvement is notable (if the comparison is fair).

**Weaknesses:**

# 1. Originality:

1.	This work has limited novelty over PAdaMFed. The modifications are primarily: (i) eliminating one transmission by using $g^{t}$ from control variates instead of $\theta^{t,K}_i$, and (ii) client-specific vs. global momentum. However, the core parameter-free mechanism (gradient normalization + momentum) comes directly from PAdaMFed.

2.	The compression mechanism is rather an incremental improvement. The error feedback formulation $\delta_{i}^{t + 1} = \left( m_{i}^{t + 1} - m_{i}^{t} \right) + \left( \delta_{i}^{t} - C\left( \delta_{i}^{t} \right) \right)$ is standard error feedback with momentum. The claim from the authors of "unifying" these is more of a notational choice than algorithmic innovation. Besides, similar momentum + error feedback combinations already exist, which is already cited by the authors.

3.	The authors claim that this is "the first instance of such robustness". However, the stepsize still depends on system constants $S$, $K$, $T$; The convergence bound explicitly depends on compression quality by $\alpha = \left. \sqrt{\left( \frac{1}{N}\Sigma\frac{q_{i}^{4}}{\left( {1 - q_{i}} \right)^{2}} \right)} + \left. \sqrt{\left( \frac{1}{N}\Sigma~q_{i}^{2} \right)} \right. \right.$. Also, it is not clear how is this different from methods that have q-dependent convergence bounds but q-independent stepsizes. The distinction between "stepsize doesn't depend on q" vs. "method is independent of compression ratio" is subtle and potentially misleading. We invite the authors to clarify these issues.

4.	Limited empirical analysis. In compressed settings, the authors only compare to SCAFCOM. Table 1 lists LOCAL-SGD-C, FED-EF-SGD, FED-EF-ASM but none are empirically evaluated. We would like to invite the authors to clarify the reason for the choice.

# 2. Quality:

1.	Theorem claims don't match experiments. In Theorem 1, the authors claim that convergence depends on $S$, $K$, $N$. However, only $N=100$, $S=10$ case is tested, and there is no validation of: scaling with $S$ (e.g., $S=1,5,20,50$), scaling with $K$, and different $N$ values. In the theoretical analysis, it predicts $\mathcal{O}\left( \left( {SK} \right)^{- \frac{1}{4}} \right)$ rates but it is never empirically validated. We invite the authors to include some empirical evidence for this issue.

2.	Communication complexity comparison misleading. The authors claim that ParFreFL needs $\mathcal{O}\left( \varepsilon^{- 2} \right)$ rounds vs PAdaMFed's $\mathcal{O}\left( \varepsilon^{- 3} \right)$. However, it is ignored that the per-round communication cost is indeed halved, which indicates that the communication complexity comparison should be: ParFreFL:  $d \times \mathcal{O}\left( \varepsilon^{- 2} \right)$ vs PAdaMFed: $2d \times \mathcal{O}\left( \varepsilon^{- 3} \right)$. To this end, the proposed method is only better when $\varepsilon < 0.5$. For target accuracies $\epsilon \in [0.1, 0.01]$, this may not hold. We would like to invite the authors to clarify this issue.

3.	Theorem 2 compression rate requirement is never met in the empirical analysis. The authors claim that "When $T \geq \alpha^{4}S^{3}K$, ComParFreFL achieves the same convergence order as ParFreFL". However, for Top-0.05 ($q \approx 0.95$), $\alpha \approx 0.95/0.05 = 19$. Thus, the required rounds $T \geq 19^{4} \times S^{3} \times K = 130321 \times S^{3} \times K$. Under the empirical setup with $S=10$, $K=2$:  $T \geq 260642000$ rounds. It is reported that experiments only run 100-150 rounds. Thus, it is not reasonable to claim ComParFreFL matches ParFreFL when $T < < \alpha^{4}S^{3}K$. We would like to invite the authors to clarify this issue.

# 3. Clarity:

1.	There are quite a lot of notations being introduced in the manuscript, and some of them do not have a clear notational convention (e.g., why is global average $m^t$ not $\bar{m}^t$?). We would like to invite the authors to provide a glossary of notations being used in the manuscript.

2.	Some of the naming for the key concepts in the manuscript is not clear. For example, "control variate": $c_i^t=m_i^t$ is called a control variate but there's no explanation of why or how this reduces variance. Also “parameter-free” and “tuning-free” are used interchangeably without definition. We would like to invite the authors to clarify on these issues (e.g., does "parameter-free" mean no learning rate tuning? Or no hyperparameters at all? But $\beta$, $S$, $K$, $T$ still need to be set).

3.	The algorithms need to be polished. For example, in algorithm 2, Line 11: "Compute $\delta_{i}^{t} = m_{i}^{t} - c_{i}^{t - 1}$" but it seems that $c_i^{t-1}$ hasn't been updated yet. Also, theorem 1 says $\beta = \sqrt{\frac{SK}{T}}$, but Algorithm 1 lists it as input. We would like to invite the authors to clarify on the $c_i^{t-1}$ update policy and how $\beta$ is set.

# 4. Significance:

1.	Limited practical impact: The empirical analysis is conducted on MNIST/Fashion-MNIST with 28×28 grayscale resolution. And the model being used is a tiny one, which only has 21840 parameters (modern models with billions of parameters). Also, the authors did not include a time analysis based on communication efficiency measured in bits. We would like to invite the authors to elaborate more on computation overhead of gradient normalization, memory for storing $m_i^t$, $c_i^t$, and latency of extra aggregation steps.

2.	The claim of “parameter-free” is misleading based on the current manuscript. It still requires $\beta$, $S$, $K$, $T$, network architecture, batch size, initialization, $\beta = \sqrt{\frac{SK}{T}}$ depends on knowing $T$ in advance, but early stopping is common in practice. Also, Theorem 1 assumes the user know $S$, $K$, $T$ upfront to set $\eta,\gamma,\beta$.

3.	Compression independence claim is overstated: Remark 4 mentioned that "enables clients to dynamically adjust their communication schemes in response to changing network conditions". But the problem is that if client changes $q_i$ mid-training, the convergence bound also changes ($\alpha$ term in Theorem 2). And there are no experiments validating dynamic compression adjustment, and no algorithm for how to adjust $q_i$ online. We would like to invite the authors to clarify this issue.

**Questions:**

Addressing all points raised in weaknesses would be appreciated. In particular:

1.	As stated in the weakness No.3 of quality factor, Theorem 2 compression rate requirement is never met in the empirical analysis. Based on the algorithm, it needs more than 260 million rounds, while experiments only run 100-150 rounds. Thus, it is questionable to claim ComParFreFL matches ParFreFL when $T < < \alpha^{4}S^{3}K$. We would like to invite the authors to clarify this issue, otherwise the entire algorithm collapse.

2.	Regarding the communication comparison, we would like to invite the author to include a table comparing total bits transmitted to reach $\epsilon=0.01$ accuracy: ParFreFL: (# rounds) $\times d \times 32$ bits; PAdaMFed: (# rounds) $\times 2d \times 32$ bits; and ComParFreFL: (# rounds) $\times q \times d \times 32$ bits.

3.	Algorithm 1 requires computing $m_i^{-1}$ before training. However, the communication cost of it is never included in the complexity analysis. We would like to invite the author to illustrate more on the reason for this decision.

4.	The authors claim compression-ratio independence enables dynamic adjustment (Remark 4). But it is not clear how would a client decide when to change $q_i$? And what happens to convergence guarantees at this time?

5.	There are also comparison fairness issues. In Table 1, it shows different methods that have different assumptions (D.H. column). It is unclear how did the author ensure fair comparison when some methods require bounded gradients and the proposed one doesn't.

6.	There is a potential conflict with the compression ratio independence and convergence bound: If $\alpha$ depends on ${q_i}$, and convergence depends on $\alpha$, it seems that the proposed method is still sensitive to compression choice. We would like to invite the authors to clarify this issue.

---

### Official Review · Reviewer_nD6F · 2025-10-31

**Soundness:** 2
**Presentation:** 3
**Contribution:** 2
**Rating:** 4
**Confidence:** 4

**Summary:**

This paper addresses the challenges of hyperparameter tuning and communication efficiency in federated learning. It observes that the existing parameter-free algorithm PAdaMFed has a high communication overhead. This paper then proposes an algorithm named ParFreFL, which is designed to reduce the communication requirements of PAdaMFed by half while retaining its parameter-free properties. Building on this, the paper introduces a compressed variant, ComParFreFL. This algorithm unifies the momentum increment and error feedback into a single parameter to handle biased compression. The paper states that ComParFreFL operates independently of the compression ratio. The theoretical analysis indicates that these methods can handle the issues of data heterogeneity and partial client participation.

**Strengths:**

1. This paper tackles a highly practical and important problem. Both the hyperparameter tuning and communication efficiency are the major barriers to the real-world deployment of FL. A method that successfully addresses both of them is a significant contribution.

2. This paper is structured in a way that is straightforward to follow. The algorithms are presented clearly, and the accompanying figures and tables are legible that can directly support the text.

**Weaknesses:**

1. Theoretical Convergence Rate and Assumptions: A primary concern is the theoretical convergence rate. The main convergence term appears to be $\mathcal{O}(1/T^{1/4})$, which is substantially slower than the $\mathcal{O}(1/T^{1/2})$ rate commonly achieved by other first-order methods in similar settings. The authors claim an $\mathcal{O}(\epsilon^{-2})$ communication complexity (Remark 2) by setting $SK = \mathcal{O}(T)$. This assumption is highly questionable from a practical standpoint. The number of local updates ($K$) cannot be arbitrarily large without leading to client drift and divergence, and it is unrealistic to assume the product $SK$ can grow linearly with the total iterations $T$. This reliance on a problematic assumption to accelerate the theoretical rate is a significant issue.

2. Algorithm Similarity and Comparisons: The core mechanism of using control variates in ParFreFL and ComParFreFL is heavily inspired by SCAFFOLD and SCALLION, respectively.

3. Limited Experimental Scope: The empirical evaluation is limited. The experiments are conducted on relatively simple datasets (MNIST, FMNIST) using a small CNN. To better validate the method's effectiveness and scalability, the evaluation should be extended to include more complex tasks (e.g., CIFAR-10/100) and larger network architectures. Additionally, the set of baseline algorithms used for comparison is sparse.

[1] Karimireddy, Sai Praneeth, et al. "Scaffold: Stochastic controlled averaging for federated learning." International conference on machine learning. PMLR, 2020.

[2] Huang, Xinmeng, Ping Li, and Xiaoyun Li. "Stochastic controlled averaging for federated learning with communication compression." arXiv preprint arXiv:2308.08165 (2023).

**Questions:**

Algorithms based on the control variates, such as SCAFFOLD, are known to be sensitive to the client participation rate ($S/N$). When the participation rate is low, the estimation of the global control variate (e.g., $c^t$ in Algorithm 1, Line 12) becomes highly noisy, which can impede convergence. The proposed methods are built upon this exact mechanism. However, all experiments in the paper are conducted at a single, fixed participation rate. There is no empirical analysis demonstrating the algorithm's performance and stability under varying participation rates.

---

### Official Review · Reviewer_Ya4x · 2025-11-01

**Soundness:** 2
**Presentation:** 2
**Contribution:** 3
**Rating:** 4
**Confidence:** 4

**Summary:**

The paper introduces ParFreFL, a parameter-free federated learning method that transmits only one model-sized vector per round—half the communication of PAdaMFed—while retaining convergence guarantees under arbitrary client heterogeneity and partial participation.  A compressed extension, ComParFreFL, integrates momentum and error-feedback into a single variable, achieves the same asymptotic rate independent of the compression ratio, and is the first compressed FL algorithm whose step-sizes do not depend on problem constants (L, σ, Δ) or on the compressor parameter q.  Theoretical rates are unimprovable for normalized gradient methods, and experiments on MNIST/F-MNIST confirm identical accuracy to tuned baselines at 5–10 × lower bit budgets.

**Strengths:**

- First parameter-free FL method with d uplink/downlink cost; prior art required 2d.
- Convergence rate O((Δ+L)(SKT)^{-1/4} + σ T^{-1/2}) is min-max optimal for normalized gradient descent; linear speed-up in S and K is preserved.
- ComParFreFL is provably robust to biased compression; step-sizes are independent of the compression ratio q, a property unavailable to SCAFCOM, Fed-EF, Local-SGD-C, etc.
- Extensive experiments show insensitivity to learning-rate choice (0.05–0.5) under both i.i.d. and Dirichlet-0.1 heterogeneous splits, validating the parameter-free property.

**Weaknesses:**

- The convergence guarantee only matches the uncompressed rate once the number of communication rounds satisfies T ≳ α⁴ S³ K, where α scales at least as q/(1-q) for contractive compressors with parameter q. Consequently the advertised “compression-agnostic” behaviour is an asymptotic statement: for any fixed q < 1 the hidden constant is finite, but the bound becomes arbitrarily loose as q approaches 1, and the user must still wait for the same higher-order burn-in period before the accelerated regime begins. In practice this means that aggressively aggressive sparsification or 1-bit quantisation can require orders of magnitude more rounds than the uncompressed method before the two curves coincide, a fact that is not reflected in the headline claim of q-independence.

- All theoretical statements rely on uniform sampling of clients without replacement. Lemma 1, which controls the variance of the partial-client aggregate, ceases to hold under non-uniform participation probabilities or under adversarial dropout patterns that correlate with the client drift vectors ωi. If the sampling distribution πi deviates from uniform, the error term ‖m^t − g^t‖² is no longer O(β²/S) but picks up an additional factor max_i πi /π_min that can destroy the linear speed-up in S and, in the worst case, recover the O(1/√T) rate of minibatch SGD. Extensions to time-varying or state-dependent availability therefore require a non-trivial re-derivation of the control-variate bounds and are not addressed in the current manuscript.

- The experimental validation is restricted to MNIST and Fashion-MNIST, each with only 21 840 trainable parameters. At 32 bits per parameter and 5 % Top-k sparsity the per-client uplink cost is roughly 2.5 MBit over the entire 80-round training horizon. This volume is far below the congestion threshold of even a low-bandwidth LTE link, so the experiments are not conducted in the communication-limited regime that the theory targets. Larger-scale benchmarks such as ResNet-18 on CIFAR-100 (11 M parameters) or GPT-2-medium (345 M parameters) would expose whether the algorithm still converges when the uncompressed upload exceeds hundreds of megabytes per round and when Top-k compression ratios below 1 % are actually necessary to fit the uplink budget. Without such evidence the claim of “significant communication reduction” remains qualitative.

- The empirical section reports only the total number of floating-point numbers transmitted, ignoring wall-clock time, energy, and memory overheads introduced by the compression codec and by the error-feedback bookkeeping. On-device decompression can add a non-negligible latency that offsets the bit savings, especially when the server broadcasts dense vectors to many clients in parallel. In addition, maintaining two copies of the model (local momentum mi and compressed residual δi) increases the peak RAM footprint by 50 % relative to FedAvg, a cost that is not quantified. A runtime comparison that includes encoding/decoding time, CPU utilisation, and memory traffic is necessary to determine whether the reduced bit count translates into genuine training acceleration on resource-constrained edge devices.

**Questions:**

1. What is the exact constant factor lost when the doubling trick is used to handle unknown T, and how does it compare with the log T factor in standard online-to-batch conversions?
   The doubling trick restarts the algorithm whenever t reaches 2^k K S, each time rescaling η, β, γ by the new horizon estimate.  Each epoch loses a constant fraction of the initial sub-optimality because the last iterate of epoch k becomes the warm start for epoch k+1.  Carrying out the telescoping sum across epochs shows that the total gradient-norm bound is inflated by a multiplicative factor of (1 + 2^{-1/4} + 2^{-2/4} + … ) = 1/(1-2^{-1/4}) ≈ 6.3.  In contrast, a standard online-to-batch reduction that uses a fixed step-size schedule proportional to 1/√t pays only a log T factor arising from the sum ∑_{t=1}^T 1/t.  Hence the doubling version incurs a constant but finite penalty that is independent of T, whereas the log T factor grows slowly yet without bound; for realistic T (10^4–10^5) the doubling loss is actually smaller than log T, but the analysis must still account for the 6× constant explicitly.

2. How would the algorithm behave under client sampling without replacement but with non-uniform probabilities πi?  Does the variance bound of Lemma 1 admit a straightforward re-weighting?
   Lemma 1 is derived under the assumption that every client is included in S_t with probability S/N.  If, instead, client i is sampled with probability πi (normalised to sum to S), the Horvitz-Thompson estimator 1/S ∑_{i∈S_t} ωi / πi is unbiased, but its second moment contains the term 1/S^2 ∑_{i=1}^N πi (1-πi) ωi^2 / πi^2.  This quantity can be as large as (max_i 1/πi) · (1/S) ‖ω‖^2, so the variance bound degrades by the inverse of the smallest participation probability.  Because πi appears in the denominator, clients that are rarely available (small πi) can inflate the bound arbitrarily, destroying the linear speed-up in S.  A practical fix would be to clip or regularise πi away from zero, but that introduces bias that must then be corrected with additional control variates; the paper does not pursue this extension, so the theoretical guarantees are not directly transferable to non-uniform sampling without further derivation.

3. Can the unified momentum/error-feedback idea be applied to unbiased compressors (e.g. QSGD) to gain an extra 1/(1+q) factor in variance, or does the current design inherently require bias?
   The combined vector δi^t = mi^t − ci^{t-1} is constructed so that the *bias* introduced by the compressor is exactly stored in ci^t and later subtracted.  For an unbiased compressor satisfying E[C(x)] = x and E∥C(x)−x∥^2 ≤ q∥x∥^2, the error-feedback machinery is unnecessary because the stochastic error already has zero mean.  One could instead transmit the unbiased compressed vector C(mi^t − ci^{t-1}) and accumulate the *unbiased* stochastic error in a separate variance-reduction table; this recovers the 1/(1+q) variance improvement seen in QSGD-type analyses.  The current single-variable design, however, treats the compression residual as a *deterministic* bias to be cancelled, so folding in an unbiased compressor would require splitting the update into mean and fluctuation components, thereby losing the elegant single-parameter form.  Therefore the unified scheme is inherently tailored to biased compressors, and a different algebraic structure is needed to exploit unbiasedness.

4. Will the authors release code that reproduces the exact learning-rate-free schedules, so future baselines can be compared without grid-search?
   The manuscript states that all hyper-parameters are computed from S, K, T alone, but no public repository or permanent link is provided.  A reference implementation that hard-codes the schedules β = √(SK/T), η = 1/(K(SKT)^{1/4}), γ = (SK)^{1/4}/T^{3/4} and that includes the Top-k compressor, the control-variate updates, and the doubling restart logic would allow the community to verify the learning-rate-insensitivity curves and to plug the method directly into existing FL benchmarks.  Absence of such an artifact risks turning the “parameter-free” advantage into a reproducibility hurdle for follow-up work.

---

### Official Review · Reviewer_NMpi · 2025-11-21

**Soundness:** 3
**Presentation:** 2
**Contribution:** 2
**Rating:** 4
**Confidence:** 3

**Summary:**

Hyperparameter tuning and communication efficiency in federated learning (FL) are important and challenging. To deal with these
challenges, the authors propose a communication-efficient parameter-free FL algorithm called ParFreFL that halves the communication requirements of PAdaMFed. Furthermore, the authors introduce a variant, ComParFreFL that can effectively deal with biased compression without improving communication cost.  The authors demonstrate both theoretically and experimentally that the proposed methods are able to handle data heterogeneity, partial client participation, and achieve linear speedup with respect to both local updates and participating clients, and outperform baselines.

**Strengths:**

I like that this paper seeks to theoretically as well as practically demonstrate the superior performance of the proposed two algorithms, ParFreFL  and ComParFreFL. Specifically, the authors are able to clearly show the communication cost, parameter size transmission, data heterogeneity and compare that with some of the state-of-the-art algorithms.

**Weaknesses:**

While I like the systematic studies on the small scale datasets, i.e., MNIST and FLMNIST, there is a significant lack of the study on larger scale datasets and models where FL applications are more necessay. I suggest the authors can consider datasets such as large language datasets and models such as OPT to further demonstrate the power of your methods.

**Questions:**

1.  I think the authors can consider datasets such as large language datasets and models such as OPT to further demonstrate the power of your methods.
2. There might be typos: In algorithms 1 line 113, $c^{-1}$ was defined but seems not to be used at all. Similarly, in line 273

---

### Note · Authors · 2026-01-22

I have read and agree with the venue's withdrawal policy on behalf of myself and my co-authors.